1

2

3

24

25 26

# Earthquake safety analysis of masonry historical building case study: Historical Konya Gazi High School

4 M. Sami Donduren<sup>1</sup>, Seyit Uguz<sup>2</sup>

5 6 <sup>1</sup>Selcuk University, Faculty of Engineering, Departmen of civil Engineering, 42060, Konya, Turkey

<sup>2</sup>Uludağ Üniversitesi, Departmen of Biyosystem Engineering, 06150, Bursa, Turkey 7

8 Correspondence to: M.Sami Donduren (sdonduren@hotmail.com) 9

10 Abstract. It is substantially significant to protect historical structures, which are an important part of our culture. 11 against natural disasters such as earthquakes and to be transmitted to future generations. The structural behaviour 12 of historical buildings must be well known to protect such structures. In order to be able to determine how safe 13 the historic buildings are against the earthquake effect, it is necessary to determine the earthquake performance. Nowadays, the most commonly used method for the modelling and structural analysis of historical buildings 14 15 systems with complex geometries is the finite element method.

16 In this study, Historical Konya Gazi High School was examined according to the present situation regarding the design and construction features with "Regulations on buildings to be built in earthquake regions" and structural 17 18 analysis was performed in ETABS program. Graphs showing displacements, moments, shear forces and axial 19 forces are used to interpret the results of the finite element analysis of the Historical Gazi High School. It has 20 been informed about the stresses and damages that may be caused by any earthquake to this building, which has 21 been serving the students for 97 years. It is aimed that this work will be a study to suggest a solution in terms of 22 not losing the our historical values and delivering it to future generations. 23

Keywords: Earthquake, finite element methods, historical buildings

## 1. Introduction

27 Earthquakes cause damages and loss of lives in urban centres and cause significant losses in rural areas as well. 28 Almost all of the buildings in the countryside, and also a large part of the old buildings in the city centre are 29 masonry buildings. In addition, many of the historical buildings were built as masonry, wood and a mixture of 30 them. There is no regulation that can be used in analysing the structural systems of such buildings. Today, 31 because of the regulations used in the design of masonry buildings are prepared for new structures, it is 32 substantially difficult to use these regulations in the study of historical structures (Akgundüz, 2004).

33 Analysis of masonry buildings is rather exhausting compared to reinforced buildings (Aköz, 2008). Analysis 34 made by package programmes for these kinds of buildings is in adequate. In recent years, through the use of 35 computer technology, plastic analysis method which the nonlinear material properties and joints are taken into 36 consideration has become more and more widely used from the classical analysis methods on the analysis of 37 masonry structures (Anonim, 2016a). There are two types of approaches in the modelling of masonry structures; 38 micro modelling and macro modelling. In the micro modelling, masonry units composed of bricks and mortar 39 are modelled by separately (Anonim, 2016b). Therefore, in the micro modelling, the mechanical properties of the 40 materials and binding materials of the structure need to be known exactly (Anonim, 2016c). Micro modelling, 41 which usually involves a large computational load, is suitable for local analysis, but is not preferred for large-42 scale analysis. (Dabanli, 2008). Applications in this model are done by using finite elements, discrete elements 43 and limit analysis. In the macro model used for plastic analysis, the mechanical material properties of structure 44 are defined by assuming as if the masonry structure materials are homogeneous (Cakti et al., 2013). The finite 45 element method is generally used in the structural analysis of masonry structures (Artar, M., 2006). Studies 46 related to earthquake were conducted by other researchers such as those given (Jeen H. W., 2017, Stephanie L., 47 2017) in Refs. In this analytical method, the structure is modelled and analysed by separating it into finite 48 elements in an appropriate number with regard to purpose of analysis. Package programs such as ETABS and 49 SAP2000 are widely used for the structural analysis done by using finite element methods.

50 In this study, the earthquake safety of the historical Konya Gazi High School was investigated according to the 51 present situation. This article provides information about the stresses and damages that may reveal due to any 52 earthquake in this building which has been serving the students for 97 years. So that, this study will suggest 53 about protection of our historical values and delivering them to future generations. Earthquake safety of the building was investigated by the ETABS programme which is one of the computer programmes used for 54 55 nonlinear static analysis. The ETABS program is software of the CSI Company and is especially designed for 56 3D static analysis of buildings. Structural analysis is done by using finite element method in the program. (Uguz, 57 2016)

1 2. Material and methods

#### 2 2.1. Information about the building

- The architecture of Konya Gazi High School, which is the subject of this study, is Mimar Muzaffer. The
- construction of the building started in 1914 and was completed in 1917. The building, which was opened in
- 1917, was used as Military High School until 1923. It was used as "Dar'ül Muallim" between 1923-1934,
- "Konya Idadi" between 1934 and 1972 and Konya High School until 1972. The layout plan of Gazi High School 7 is given in Figure 1.

9

Figure 1. The layout plan of Gazi High School

## 10 2.2. Architectural features

11 The historical Gazi High School is located in Konya city center, at the intersection of Atatürk Street and Amber 12 Reis Street. The building is positioned the south of the school area. There are other buildings for sports hall, 13 laboratories and conference halls in the school garden. The empty space in the middle of these three buildings is 14 used as a sports and ceremonial space. Because the building is a historical building, there is no architectural or 15 static project of the building. For this reason, the architectural project of the building was made by taking the 16 relievo. Konya Gazi high school has a basement floor, ground floor and two normal floors. The height of the 17 floors differs from floor to floor. Basement and ground floor heights are 5,00 m, first floor and second heights 18 are 4,50 m.

### 19 2.3. Structural System and Material Properties

20 The building is not exactly symmetric and also is built with masonry structural system. The form of the 21 structural system varies with each floor. It was observed that rubble stone was used as material in the walls. It is 22 thought that the rubble stones used in this structure are brought from the Sille region in Konya. When the walls 23 of the structural system elements are examined, it is observed that the basement wall thickness is 90 cm, the 24 ground floor wall thickness is 80 cm, the first floor wall thickness is 75 cm and the second floor wall thickness is 25  $\overline{70}$  cm. It is known that the second floor of the building was rebuilt with renovation, but it could not be verified because there were not enough resources. . The basement floor, the ground floor and the second floor slabs are 26 27 not visible from the coatings. However, it has been observed that horizontal beams were used in the first story. 28 Figure 2 shows the image of the first floor slab.

Figure 2. Horizontal beams at the first floor

## 3. Analysis programme

The ETABS program is one of the computer programs used for nonlinear static analysis. The program is software of CSI Company with ISO9001 quality certification and is specially designed for 3D static analysis of

33 34 building type structures.

35 The CSI Company was founded in 1975 and is the manufacturer of programs, which are used in more than 160 36 countries worldwide. This program is also used in project designs of buildings such as Taipei Finance Centre in

© Author(s) 2018. CC BY 4.0 License.

- Taiwan, One World Trade Centre in New York and Beijing National Stadium. ETABS program analyses by 1
- using the finite element method (Sırlıbaş, 2013).
- 3.1. Modelling and analysis in ETABS 2015 program
- Structural Modelling
- In order to assessment the earthquake performance of the building, the Regulation on Buildings to be done in
- Earthquake Regions (DBYBHY, 2007) was followed. However, FEMA 356 (Prestandard and Commentary for
- Seismic Rehabilitation of Buildings) regulation is used in cases where our current earthquake regulations may be
- insufficient. First of all, axes were determined on the floor plans of the building while modelling was doing.
- Figure 3 shows the axes of the floor plan in the AutoCAD program.

11

Figure 3. Axes in the basement floor plan

Afterwards axes were determined in the ETABS program and the elements coming from those axes were 14 modelled. In literature, macro and micro finite element models can be shown as advanced modelling techniques 15 of masonry structures. In the scope of this study, the structural analysis was carried out with the macro model of 16 the finite element method created by ETABS 2015 software. The front and rear front of the building is shown in

17 Figure 4.