# Peer review of "Earthquake safety analysis of masonry historical building case study: Historical Konya Gazi High School"

_Natural Hazards and Earth System Sciences, 2017_

## Referee Comment (RC1) · Anonymous Referee #1 · 15 Feb 2018

In this manuscript, the main objective is to assess the seismic behaviour of an historical existent masonry building. Nevertheless, according to this review, the scientific and technical content of the paper is not appropriate to be publish in an International Journal; thus, it is not adequate for the NHESS Journal. The paper deals with a subject interesting and worth to be studied. However, there are several weaknesses that could not be accepted.

In the following will be listed the main comments/corrections to be addressed: • the English need a careful and deep revision. A huge number of grammar errors exist and some sentences should be completely re-written. It is suggested that the paper

needs revision by native speakers  ́c There are several parts that are not expected to be added in this type of work (just one example among many: page two, lines 35-36..'' The CSI Company was founded in 1975 and is the manufacturer of programs, which are used in more than 160 countries worldwide. This program is also used in project designs of buildings such as Taipei Finance Centre in. . ..'').  ́c There several parts missing and are crucial for the adequate seismic assessment of an old masonry buildings. For instance: despite the modal analysis performed for the dynamic characterization of the building, what type of analysed was used? Only a linear equivalent seismic load (i.e. a linear static analysis)? Is it adequate for adequate for the seismic assessment of an old masonry building?  ́c The results and the discussion of the results are unappropriated.

There are minor comments that could also be added:  ́c The references need revision (for the programs a revision should be added, etc.)  ́c The quality of some figures needs improvement (e.g. Figure 6)  ́c In English the decimal numbers are identified with dots and not with commas..

---

## Referee Comment (RC2) · M. Kamanly (Referee) · 19 Feb 2018

This paper is about the earthquake damage on the historical building. this paper is well organised and also contributes to the science.

---

## Referee Comment (RC3) · Anonymous Referee #3 · 20 Feb 2018

The paper in consideration is intituled "Earthquake safety analysis of masonry historical building case study: Historical Konya Gazi High School". The main purpose was to proceed to a numerical analysis of a historical masonry building, Konya Gazi High School, in Turkey, in order to evaluate the structural seismic performance. In term of scientific contents, the importance and relevance of studies addressing structural assessment of existing building subject to seismic events are evident. In general, the text needs significant editing by a native speaker. Even, in some parts of the text, namely, in tables and figures, the text is not in English. In terms of technical contents, the different modelling approaches in masonry buildings (micro/macro modelling), the differentiation between static/dynamic analyses types, the assumption of a linear and

non-linear behaviour, are addressed in the paper scope. However, the text does not completely clarify what type of analysis was adopted. The explanation of the options taken is essential to fully understand the work presented. From a perspective of the practical application, as mentioned by the authors, the reference to recommendation and standards concerning assessment of existing buildings in seismic risk zones is crucial. The authors point out some criteria defined in the FEMA 356 - Prestandard and Commentary for the Seismic Rehabilitation of Buildings. However, in the opinion of this reviewer, the analysis of the results could be further explored and analysed, namely, in terms of standard limits. A solution to improve the observed structural behaviour should be also proposed and analyse. Furthermore, the comparison with the European standard - Eurocode 8: Design of structures for earthquake resistance - Part 3: Assessment and retrofitting of buildings - could enrich the work presented.

---

## Author Comment (AC1) · 3 May 2018

Referee 1 We thank Reviewer 1 for your helpful, thoughtful comments and have made a lot reversion about the manuscript following the suggestion. We agree with almost all your comments and we have revised our manuscript accordingly. Revisions belonging to the Referee 1 are marked with yellow colour, and revisions belonging to Referee 3 are marked with red colour in the text. 1. The English need a careful and deep revision. A huge number of grammar errors exist and some sentences should be completely re-written. It is suggested that the paper needs revision by native speakers Response: The manuscript has been edited by an English-speaking native, so we

hope it now matches the journal standard. 2. There are several parts that are not expected to be added in this type of work (just one example among many: page two, lines 35- 36..." The CSI Company was founded in 1975 and is the manufacturer of programs, which are used in more than 160 countries worldwide. This program is also used in project designs of buildings such as Taipei Finance Centre in. . ..."). Response: These parts were also changed. 3. There several parts missing and are crucial for the adequate seismic assessment of an old masonry buildings. For instance: despite the modal analysis performed for the dynamic characterization of the building, what type of analysed was used? Only a linear equivalent seismic load (i.e. a linear static analysis)? Is it adequate for the seismic assessment of an old masonry building? Response: The missing parts which wasn't mention about analysis was also changed in the full text. "Structural analysis for Finite Element Model of the building is done with linear analyse by using ETABS program. The seismic analysis of the structure studied in this article, is done by using Equivalent Earthquake Load Method as described in the Turkish Codes-2007. Mode shapes of the building have been obtained by modal analysis approach using ETABS program. Modal analysis was performed in 12 modes with Eigen Vectors to determine free vibration periods and mode shapes of the building." 4. The results and the discussion of the ′ results are unappropriated Response: The results and the discussion was also editted in the study. In addition to this comments, the references has been revised. There were some words in Turkish in the figures. They are also changed. Decimal numbers are identified with dots.

Please also note the supplement to this comment:
https://www.nat-hazards-earth-syst-sci-discuss.net/nhess-2017-449/nhess-2017-449-AC1-supplement.pdf

---

## Author Comment (AC2) · 3 May 2018

**Earthquake Safety Analysis of Masonry Historical Building 1 2 Case Study: Historical Konya Gazi High School 3 4 M. Sami Donduren1, Seyit Uguz2 5 6 7 Selcuk University, Faculty of Engineering, Department of Civil Engineering, 42060, Konya, Turkey 8 Uludag University, Department of Biosystems Engineering, 16150, Bursa, Turkey 9 10 Abstract. SUMMARY 11 It is substantially significant to protect insulate historical structures, which are an important part of our culture, 12 against for natural disasters such as earthquakes and to be transmitted to future generations. The structural 13 behaviour of historical buildings must beis difficult to well knowncharacterized to protect such structures 14 order to be able to determine how safe the historic buildings are against the earthquake effect, it is necessary to 15 determine the earthquake performance of the historical buildings in order to determine how safe the historical 16 buildings are for the earthquake effect. Nowadays, the most commonly used method for the modelling and 17 structural analysis of historical buildings systems with complex geometries is the finite element method. 18 In this study, Historical Konya Gazi High School was examined according to the present situation regarding the 19 design and construction features with "Regulations on buildings to be built in earthquake regions" and structural 20 analysis was performed in ETABS program. Graphs showing displacements, moments, shear forces and axial 21 forces are used to interpret the results of the finite element analysis of the Historical Gazi High School. It has 22 been informed about the stresses and damages that may be caused by any earthquake to this building, which has 23 been serving the students for 97 years. It is aimed that this work will be a study to suggest a solution in terms of 24 not losing the ourour historical values and delivering it to future generations. 25 Keywords: Earthquake, finite element methods, historical buildings 26 1. INTRODUCTION 27 28 Earthquakes cause damages and loss of lives in urban centres and cause significant losses in rural areas as well. 29 Almost all of the whole buildings in the countryside, and also a large part of the old buildings in the city centre 30 are masonry buildings. In addition, many of the historical buildings were built as masonry, wood and a mixture 31 of them. There is no regulation that can be used in analysing the structural systems of such buildings. Today, 32 regulations used in the design of masonry buildings are prepared for new structures, it is of the 33 substantially difficult to use these current regulations in the study of historical structures -because of these 34 regulations used in the design of masonry buildings are prepared for new structures. 35 Analysis of masonry buildings is rather exhausting complicated compared to reinforced buildings. Analysis 36 made by package Package programmes used to analyse for these kindkindse of buildings is are in-adequate. In 37 recent years, through the use of computer technology, plastic analysis method, -which the nonlinear material 38 properties and joints are taken into consideration, has become more and more widely used from than the classical 39 analysis methods on the analysis of for masonry structures. There are two types of approaches in the modelling 40 of masonry structures; micro modelling and macro modelling. In the micro modelling, masonry Masonry units 41 composed of bricks and mortar are modelled by separately in the micro modelling. Therefore, in the micro 42 modelling, the mechanical properties of the materials and binding materials of the structure need to be known**

Biçimlendirilmiş: Yazı tipi: 17 nk Biçimlendirilmiş: Sola

| Biçimlendirilmiş:                 | Yazı tipi: | Kalın Değ |
|-----------------------------------|------------|-----------|
| Biçimlendirilmiş:                 | Üst simge  | e         |
| Biçimlendirilmiş:                 | Yazı tipi: | Kalın Değ |
| Biçimlendirilmiş:                 | Üst simge  | e         |
| Biçimlendirilmiş:
simge | Yazı tipi: | 10 nk, Üs |
| Biçimlendirilmiş:                 | Yazı tipi: | 10 nk     |
| Biçimlendirilmiş:
simge | Yazı tipi: | 10 nk, Üs |
| Biçimlendirilmiş:                 | Yazı tipi: | 10 nk     |
| Biçimlendirilmiş:
satır        | Satır aral | ığı: 1,5  |

Biçimlendirilmiş: Yazı tipi: 10 nk Biçimlendirilmiş: Satır aralığı: 1,5 satır

characterized exactly. Micro modelling, which usually involves a large computational load, is suitable for local 43 44 analysis, but is not preferred for large-scale analysis. (Dabanli, 2008). Applications in this model are done by 45 using finite element methods, discrete elements and limit analysis. In the macro model used for plastic analysis, 46 the mechanical material properties of structure are defined by assuming as if the masonry structure materials are 47 homogeneous (Cakti-Cakti et al., 2013). The finite element methodFEM is generally used in the structural 48 analysis of masonry structures. In this analytical method, the structure is modelled and analysed by separating it 49 into finite elements in an appropriate number with regard to purpose of analysis. Package programs such as 50 ETABS and SAP2000 are widely used for the structural analysis done by using finite element methods.FEM. 51 In this study, the earthquake safety of the historical Konya Gazi High School was investigated according to the 52 present situation. This article provides information about the stresses and damages that may reveal due to any 53 earthquake in this building. which has been serving the students for 97 years. So that, tThis study will suggest 54 about protection of our historical values and delivering them to future generations. Earthquake safety of the 55 building was investigated by the ETABS programme which is one of the computer programmes used for 56 nonlinear static analysis. The ETABS program is software of the CSI Company and is especially designed for 57 3D static analysis of buildings. Structural analysis is done by using finite element method in the program.

58 2. CASE STUDY: HISTORICAL KONYA GAZI HIGH SCHOOL

59 2.1. Information about the buildingdentity of the Structure

The architecture of Konya Gazi High School, which is the subject of this study, is Mimar Muzaffer. The construction of the building started in 1914 and was completed in 1917. The building, which was opened in 1917, was used as Military High School until 1923. It was used as The name of the school was "Dar'ül Muallim"
between 1923-1934, "Konya Idadi" between 1934 and 1972 and Konya High School until 1972. The layout plan of Gazi High School is given in Figure 2.1.

Biçimlendirilmiş: Yazı tipi: 10 nk

Biçimlendirilmiş: Yazı tipi: 10 nk

Biçimlendirilmiş: Yazı tipi: 10 nk

Biçimlendirilmiş: Yazı tipi: 10 nk Biçimlendirilmiş: Yazı tipi: 10 nk

65 66

67

68

69

70

2.2. Architectural features

Figure\_-2.1. The layout plan of Gazi High School

The historical Gazi High School is located in Konya city centercentre, at the intersection of Atatürk Street and Amber Reis Street. The building is positioned the south of the school area. There are other schoolhouse. buildings for sports hall, laboratories and conference halls in the school garden. The observed building is positioned the school area. The empty space in the middle of the set three buildings is used as a sports and ceremonial space. Because the building is a historical building, there-There is no architectural or static project of this historical e-building. For this reason, the architectural project of the building was made by taking

the relievo. Konya Gazi high school has a basement floor, ground floor and two normal floors. The height of the floors differs from floor to floor. Basement and ground floor heights are  $5_{\frac{1}{2}}00$  m, first floor and second floor

**71 72**

73

76 heights are 4,...50 m. In general, the condition of the building is good although some of the bearing walls have

77 local cracks, deterioration of mortar or stone, roof elements.

78 2.3. Structural System and Material Properties

88

79 The building is not exactly symmetric and also is built with masonry structural system. The form of the structural 80 system varies with each floor. It was observed that rubble stone was used as material in the walls. It is thought 81 that the rubble stones used in this structure are brought from the Sille region in Konya. When the walls of the 82 structural system elements are examined, it is observed that the basement wall thickness is 90 cm, the ground 83 floor wall thickness is 80 cm, the first floor wall thickness is 75 cm and the second floor wall thickness is 70 cm. 84 It is known that the second floor of the building was rebuilt with renovation for restoration, but it could not be 85 verified because there were not enough resources.- The basement floor, the ground floor and the second floor 86 slabs are not visible from the coatings. However, it has been observed that horizontal beams were used in the 87 first story. Figure 2.2 shows the image of the first floor slab.

Biçimlendirilmiş: Yazı tipi: 10 nk

**Biçimlendirilmiş:** Girinti: İlk satır: 0 cm, Satır aralığı: 1,5 satır

Biçimlendirilmiş: Yazı tipi: 10 nk Biçimlendirilmiş: Satır aralığı: 1,5 satır

Biçimlendirilmiş: Yazı tipi: 10 nk

Biçimlendirilmiş: Üst simge

**Biçimlendirilmiş: Yazı tipi: 10 nk**

**Biçimlendirilmiş:** Satır aralığı: 1,5 satır, Numaralandırılmış anahat + Düzey:2 + Numaralandırma Stili: 1, 2, 3, ... + Başlangıç: 4 + Hizalama: Soldan + Hizalandığı yer: 0 cm + Girinti yeri: 0,63 cm

**Biçimlendirilmiş:** İki Yana Yasla, Satı aralığı: 1,5 satır

Biçimlendirilmiş: Yazı tipi: 10 nk Biçimlendirilmiş: Satır aralığı: 1,5 satır

| 89  | Figure 2.2. Horizontal beams at the first 1 st floor                                                 |
|-----|-----------------------------------------------------------------------------------------------------------------|
| 90  | 3.2.4. Structural ModellingANALYSIS PRO GRAMME                                                                  |
| 91  | The program used to static analysis of building is the software ETABS 2015, a specialist program for the three  |
| 92  | dimensional analysis and design of building systems. Finite Element Method model was formed to calculate the    |
| 93  | response of the structure with this program. The finite element mesh shows the real mass distribution for the   |
| 94  | ideal concentration of the masses at the nodes.                                                                 |
| 95  | The ETABS program is one of the computer programs used for nonlinear static analysis. The program is            |
| 96  | software of CSI Company with ISO9001 quality certification and is specially designed for 3D static analysis of  |
| 97  | building type structures.                                                                                       |
| 98  | The CSI Company was founded in 1975 and is the manufacturer of programs, which are used in more than 160        |
| 99  | countries worldwide. This program is also used in project designs of buildings such as Taipei Finance Centre in |
| 100 | Taiwan, One World Trade Centre in New York and Beijing National Stadium. ETABS program analyses by              |
| 101 | using the finite element method (Sırlıbaş, 2013).                                                               |
| 102 | 3.1. Modelling and analysis in ETABS 2015 program                                                               |
| 103 | Structural Modelling                                                                                            |
| 104 | In order to assessment the earthquake performance of the building, the DBYBHY, 2007 (The Regulations on         |
| 105 | Buildings to be built done in Earthquake Regions) (DBYBHY, 2007) was followed in order to assessment the |
| 106 | earthquake performance of the building However, FEMA 356 (Prestandard and Commentary for Seismic                |
| 107 | Rehabilitation of Buildings) regulation is used in cases where our current earthquake regulations may be        |

---

## Author Comment (AC3) · 3 May 2018

Referee 3 We thank Reviewer 3 for your helpful, thoughtful comments and have made a lot reversion about the manuscript following the suggestion. We agree with almost all your comments and we have revised our manuscript accordingly. Revisions belonging to the Referee 1 are marked with yellow colour, and revisions belonging to Referee 3 are marked with red colour in the text. • There were some missing words in Turkish in the figures and tables. They are changed with English. The manuscript has been edited by an English-speaking native, so we hope it now matches the journal standard. • The missing part which wasn't mention about analysis was also changed in the

full text. Edited text is the below; "Structural analysis for Finite Element Model of the building is done with linear analyse by using ETABS program. The seismic analysis of the structure studied in this article, is done by using Equivalent Earthquake Load Method as described in the Turkish Codes-2007. Mode shapes of the building have been obtained by modal analysis approach using ETABS program. Modal analysis was performed in 12 modes with Eigen Vectors to determine free vibration periods and mode shapes of the building." • Actually, the analysis of the results was categorized as modal analysis results, axial stress results, shear stress results and Displacement results. Limit values were compared with standard limits on conclusion in terms of Turkish Standards. As you mention, it could be better if we interpret the detailed comparisons with FEMA 356 and Eurocode 8 in this study. We will consider your these comment in our future studies.

Please also note the supplement to this comment:
https://www.nat-hazards-earth-syst-sci-discuss.net/nhess-2017-449/nhess-2017-449-AC3-supplement.pdf